# Functional identification of Annexin B1 and Annexin B2 from *Cysticercus cellulosae* and their mechanism in plasma membrane repair

**Peixia He[1◉], Dejia Zhang[2◉], Mengqi Wang[3], Rui Duan[3], Yuyuan Zhao[3], Sirui Wang[3], Xing Yang[4], Xiaolei Liu[2]\*, Shumin Sun ◉[1,3]\***

**1** College of Veterinary Medicine, Yunnan Agricultural University, Kunming, China, **2** State Key Laboratory for Diagnosis and Treatment of Severe Zoonotic Infectious Diseases, Key Laboratory for Zoonosis Research of the Ministry of Education, Institute of Zoonosis, and College of Veterinary Medicine, Jilin University, Changchun, China, **3** College of Animal Science and Technology, Inner Mongolia MinZu University, Inner Mongolia Tongliao, China, **4** Integrated Laboratory of Pathogenic Biology, College of Preclinical Medicine, Dali University, Dali, Yunnan, China

◉ These authors contributed equally to this work.
\* liuxlei@163.com (XL); shums1975@163.com (SS)

## Abstract

### Background

Cysticercosis is a severe foodborne zoonotic parasitosis infected by the metacestode larvae of *Taenia solium*. However, its invasion mechanism is still not clear, which might provide the important evidence for treatment or vaccine. It was reported the annexin involved in the physiological and pathological functions of *Cysticercus cellulosae*. However, the regulatory mechanisms and roles of annexin B1 and annexin B2 in the invasion and immune escape of *Cysticercus cellulosae* have not been fully explored.

### Methods

The annexin was acquired by cloning in prokaryotic expression vector, expressed in *Escherichia coli*, and purified by affinity chromatography. Its expression was determined by immunohistochemistry. The anticoagulant function and its underlying mechanism was verified by the determination of activated partial thromboplastin time, prothrombin time and phospholipid binding activity. The membrane repair function was verified by cell culture, transfection, and laser confocal technology.

### Results

Immunohistochemistry results showed the B1 and B2 were mainly expressed on the body surface and the surface of digestive glands of *Cysticercus cellulosae*. The Blood coagulation results illustrated the B1 and B2 can prolong the time of both exogenous and endogenous coagulation pathways, with B2 having a more significant effect. They tend to bind to phosphatidylserine, possibly interfering with coagulation complex formation and inhibiting

**Data availability statement:** All relevant data are within the paper and its Supporting information files.

**Funding:** SS prof. was supported by the National Nature Science Foundation of China (NSFC 32160842, 31960707, 32460893), the Inner Mongolia autonomous region science and technology plan project (2022YFDZ0049), the Natural Science Foundation of Inner Mongolia Autonomous Region (2021MS03037), the Basic scientific research operating expenses of colleges and universities directly under Inner Mongolia Autonomous Region project (GXKY22042); XL prof. was supported by the National Key Research and Development Program of China (2022YFE0114400, 2022YFE0114600), the National Nature Science Foundation of China (31460658). The funders had no role in study design, data collection and analysis, decision to publish, or preparation of the manuscript.

**Competing interests:** The authors have declared that no competing interests exist.

the coagulation pathway, and may assist in the worm's penetration through blood vessels and migration to parasitic sites. The plasma membrane repair test revealed the cells transfected with B1 and B2 genes have a significantly shorter plasma membrane repair time than the control group, suggesting that these proteins may be involved in repairing the worm's body surface to resist the immune system's attack when the host immune system attacks.

## Conclusions

The Annexin B1 and Annexin B2 of *Cysticercus cellulosae* possess anticoagulant properties and can assist in membrane repair. Given these functions, it is speculated that they play a crucial role in immune evasion and invasion. However, further experiments are required to provide direct evidence to further validate these speculations.

## Author summary

Cysticercosis is a significant foodborne zoonotic parasitic disease, and it is also a neglected disease. It has excellent immune evasion ability and complex evasion mechanisms (antigen simulation and camouflage, plasma membrane isolation, immunosuppression, etc.), which is one of the important reasons why the disease is still prevalent. This study investigated the possible roles of *Cysticercus cellulosae* Annexin B1 and Annexin B2 in the invasion of hosts and immune evasion by *Cysticercus cellulosae*. We found that its coagulation function showed inconsistent effects compared to previous studies. To our knowledge, this is the first discovery of an invertebrate annexin facilitating plasma membrane repair. This has opened up new ideas for the prevention and control of *Cysticercus cellulosae*, as well as for the development of drug targets. In conclusion, this study provides new perspectives and valuable clues for further elucidation of the invasion mechanism of *Cysticercus cellulosae*.

## 1. Introduction

*Taenia solium* belongs to Taeniidae, its metacestode stage is known as *Cysticercus cellulosae*. There are approximately 20 species of parasitic tapeworms in the Taeniidae, including *Taenia Saginata*, *Taenia Hydatigena*, *Taenia Pisiformis*, and *Taenia asiatica* and so on. The *Taenia solium*, *Taenia saginata*, and *Taenia asiatica* can only cause human taeniasis, among which *Taenia solium* being the most harmful. When parasitic in muscle or subcutaneous tissue, the clinical symptoms were not obvious, except for the detectable bean-sized nodules [1]. Notably, the *Cysticercus cellulosae* infection on the eyes and brain can cause serious health problems. It was reported the *Cysticercus cellulosae* under retina can cause vision loss and even blindness [1]. The clinical manifestations of neurocysticercosis are complex. Epilepsy and headaches were the most common symptoms. The uncommon symptoms include intracranial hypertension, hydrocephalus, cognitive impairment, depression, and even suicidal tendency [2,3]. It was reported *Taenia solium* prevalent in 31 countries in Africa, 16 countries in Central America and the Caribbean Basin, and 8 countries in East Asia and Southeast Asia [4–6].

One of the fundamental reasons for the epidemic of the disease is the complex mechanisms of invasion and immune escape. Exploring the invasion mechanism can provide data support and theoretical basis for blocking invasion and diagnosis. The *Cysticercus cellulosae* was

encompassed by a cystic membrane that originates from the host tissue. This cystic membrane served as a physical barrier to safeguard the worm, preventing cells and molecules from the host's immune system from having direct contact with the worm, thereby facilitating immune escape [7]. However, proteins expressed or secreted by cysticerci also play a significant role in immune evasion. 76 proteins were reported in the proteomics of excretory-secretory protein of *Cysticercus cellulosae*, of which 17 proteins originating from the host [8]. These host protein components acted as camouflage for worms, thus blocking the recognition and immunity of the host immune system towards them. The suppression of the host's immune response was one of the most typical mechanisms for parasite immune evasion. Whether *Cysticercus cellulosae* survives in the brain would determine if the patient would exhibit clinical symptoms, indicating that *Cysticercus cellulosae* can regulate the host's immune response. Calcified *Cysticercus cellulosae* triggered the production of IFN-γ, TNF-α, IL-17, and IL-13. This leads to a pro-inflammatory Th1 type response, resulting in the manifestation of clinical symptoms. However, the surviving *Cysticercus cellulosae* induced the production of IL-4, IL-5, IL-10, and IL-13, which made the immune response tend towards an anti-inflammatory Th2 response, resulting in absence of clinical symptoms [9]. Another key strategy developed by *Cysticercus cellulosae* to control innate immunity was to block the complement system by releasing antigens, such as taeniaestatin, inhibited both the classical and alternative complement pathways [10]. Additionally, paramyosin bind to complement component C1q and inhibits its activity [10]. Living cysts could secrete cysteine proteases and metalloproteinases, which can degrade host immunoglobulins, interfere with the proliferation of CD4+ cells, and inhibit the production of cytokines [11]. Due to its numerous antigens, not all of them have been thoroughly studied yet. Therefore, our understanding of the complex immune evasion mechanisms will be further enhanced in the future.

Annexin (ANX) is a calcium-dependent phospholipid-binding protein with a highly conserved sequence throughout evolution. Any member of the Annexin family can be divided into two regions: one is the highly conserved core region-C-terminal, and the other is the highly variable N-terminal [12]. Thehighly conserved C-terminal contains four highly similar annexin repeat domains, each of which was a highly α-helical structure, which was composed of about 70 amino acid residues, while the N-terminal consists of 20–200 amino acid residues [13]. The amino acid sequence differences among Annexin family members were primarily observed in the N-terminal region. Therefore, it was speculated that the different functions of each Annexin member were closely related to the amino acid sequence of the N-terminal structural region. Annexin has two fundamental biological characteristics. First, it could only bind to phospholipid membrane with the participation of $Ca^{2+}$, and Annexin mainly binds to acidic phospholipids. Secondly, this binding was reversible [14]. The binding between Annexin and the phospholipid membrane would be separated after the removal of $Ca^{2+}$ [14]. Annexin has various functions, including anti-inflammatory properties, maintaining fibrinolytic balance, anticoagulant, regulating vesicle transport, and controlling the formation of ion channels [15–19].

ANX A1 was expressed in parasitic vacuoles containing *Leishmania* and played a key role in the fusion of vesicles and endosomes [20]. This helps to elucidate the regulatory mechanism of ANX A1's cellular role during *Leishmania* infection. The expression of ANX A13 of *Schistosoma japonicum* was analyzed across different life cycle stages. It was observed that the expression level of ANX A13 varied among genders and even before and after mating. The difference in this expression pattern may indicated that *Schistosoma japonicum* played distinct roles in male and female reproductive development [21]. Rats immunized subcutaneously with ANX B30 of *Clonorchis sinensis* could induce Th1/Th2 combined immune response. The levels of IgG1 and IgG2a were significantly increased

(especially IgG1), and the levels of IL-10 and IFN-γ were slightly increased [22]. Further study of ANX B30 revealed the mechanism by affecting the host immune response during infection with *Clonorchis sinensis*. The expression of four annexins (NEX-1, 2, 3, 4) was also different in different periods of *Caenorhabditis elegans*, and their glycosaminoglycan binding activity also differed [23]. It was speculated that the four proteins play different roles in the development of the *Caenorhabditis elegans*. Annexin B1 and B2 have been reported in *Cysticercus cellulosae* [24,25]. It is currently known that Annexin B1 can bind to the extracellular surface of human eosinophils and induce their apoptosis [26]. Additionally, Annexin B1 has been shown to inhibit phospholipase A2 in mammals in vitro, which may contribute to the reduction of the host's inflammatory response [27]. Furthermore, both Annexin B1 and B2 have demonstrated anticoagulant effects in laboratory tests [24,25]. Those studies have confirmed that annexin played a crucial role in the life activities of parasites and significantly influences their parasitism, migration, growth, and development. However, research on the function of annexin in parasites was only focused on blood coagulation, parasite-host interaction, immunomodulation, and so on. The plasma membrane is a fundamental barrier that isolates the cell from its surrounding environment. The exchange of substances across the membrane is strictly regulated to maintain a stable biochemical environment, thereby ensuring the normal function of cells. Once damaged, the membrane can severely disrupt cellular functions. The repair function of annexins on the plasma membrane has been confirmed in A1, A2, A5, A6, A7, and A8 [28]. The long-term survival of *Cysticercus cellulosae* in the host and their continued infectivity are closely related to the maintenance of cystic integrity. However, whether Annexin B1 and B2 can repair the plasma membrane (encapsulation) remains to be verified.

The annexins B1 and B2 of *Cysticercus cellulosae* were associated with the function of plasma membrane repair for the first time in our study. We speculated that these two proteins played a more important role in the invasion mechanism and immune evasion of *Cysticercus cellulosae*. To address this proposal, the Annexin B1 and Annexin B2 of *Cysticercus cellulosae* were expressed, and their functions were verified. The results supported that Annexin possibly played an important role in maintaining structural integrity and promoting parasite migration and parasitism. However, further experiments are required to provide direct evidence to further validate these speculations. It provided a new perspective and valuable clues for an in-depth understanding of the invasion mechanism of *Cysticercus cellulosae*.

## 2. Materials and methods

### 2.1. Ethics approval and consent to participate

This study was approved by the ethics committee of Inner Mongolia MinZu University (approval no.IMUN20190301). We certify that the study was performed in accordance with the 1964 declaration of HELSINKI and later amendments.

All participants who were >18 years old were informed and enrolled in the research. This study was approved by the Ethical Committee of Jilin University, China (ethical clearance number # 2021703).

Written informed consent was obtained from all the participants prior for the publication of this study.

### 2.2. Statement approved by the institutional animal care and use committee

This study carried out the principles of laboratory animal affairs management in China. The experiment was approved by the Animal Protection and Use Committee of Jilin University

(20170318). All experiments were performed in accordance with relevant guidelines and regulations.

## 2.3. Study animals

Five-month-old female rabbits weighing 2.0 ± 0.2 kg were maintained under standard conventional conditions. All procedures were in strict accordance with the People's Republic of China legislation on the use and care of laboratory animals. The source of the positive serum of porcine cysticerci can be found in published articles [29].

## 2.4. Expression of Annexin B1 and Annexin B2

The gene sequences of Annexin B1 and Annexin B2 can be obtained from GenBank, with accession numbers AAD34598.1 and AY998562.1, respectively. The target gene was cloned into the pET-28a(+) prokaryotic expression vector. The recombinant plasmid was then transferred into *Escherichia coli* BL21 (DE3) cells after 42°C thermal stimulation. The recombinant plasmid was spread on a plate containing 30 µg/mL of kanamycin and cultured at 37°C. The monoclonal bacteria were cultured at 37°C in a liquid medium containing 30 µg/mL kanamycin. 0.5 mM IPTG was added when the OD value reached 0.6. Different groups were set up: one group was cultured overnight at 20°C, while the other group was cultured at 37°C for 6 h. The group without IPTG was used as the negative control. The bacterial liquid was centrifuged at 4000 rpm for 10 minutes. The supernatant was discarded, and the bacteria were collected. The collected bacteria were floated with PBS and completely dissolved on ice using an ultrasonic crusher (power 30%, working 5s, intermittent 5s). The supernatant and precipitate were collected after centrifugation, and the precipitate was dissolved by binding buffer. The supernatant and precipitated proteins were prepared respectively, and their expression was detected by SDS-PAGE.

## 2.5. Purification of Annexin B1 and Annexin B2

Culture bacteria in a medium containing 30 µg/mL of kanamycin. 0.5 mM IPTG was added when the OD value reaches 0.6. After being cultured overnight at 20°C to induce high levels of expression, the bacteria were collected by centrifugation. The bacteria were dissolved in lytic buffer, broken by ultrasound. Then the supernatant containing crude proteins was collected by centrifugation. Take 5 mL Ni-NTA and clean the balance column with b-inding buffer with 5 times column volume. The crude protein was incubated with the balanced nickel column for 1 hour, and then the eluate was collected. Cleaned and balanced the columns with binding buffer. Cleaned the Ni²⁺ affinity chromatography column with washing buffer and collected the eluate. Eluted with elution buffer, and collected the eluate. The crude protein and eluate were treated and prepared for SDS-PAGE, respectively. The purified components were dialyzed into the protein preservation buffer (50 mM Tris, 300 mM NaCl, 0.1% sarkosyl, 2 mM DTT, pH 8.0). It was concentrated with PEG2000 after dialysis, then filtered with 0.45 µm filter membrane, and determined by BCA method. Finally, it was stored at -80 °C.

## 2.6. SDS-PAGE and western blot

The target protein was separated by 12% polyacrylamide gel, stained with Coomassie brilliant blue and decolorized. The purified protein was separated by 12% polyacrylamide gel and then transferred to NC membrane. The membrane was incubated with *Cysticercus cellulosae* positive serum (1:200) or with anti His monoclonal antibody (1:3000) at 4 °C for 12 h. After washing five times, HRP-goat anti-pig or HRP-goat anti-mouse IgG was added and incubated

at room temperature for 1 h. Detection of immunoreactive proteins was performed by a chemiluminescent substrate system.

## 2.7. Polyclonal antibody production

The recombinant proteins B1 and B2 were used as immune antigens. 500 μg recombinant protein was mixed with Freund's complete adjuvant at 1:1 for emulsification. Rabbits were immunized by subcutaneous injection of more points under the back. The interval between each immunization was 15 days. 250 μg protein was mixed with Freund's incomplete adjuvant at 1:1 in the second, third, and fourth times, and the immune operation was also carried out. Blood collection was performed every 7 days through the ear vein.

The polyclonal antibody titer was determined by ELISA, and the antibody specificity was determined by Western blot. Dilute Annexin B1 and Annexin B2 to 10 μg/mL using the CBS (15 mM/L $Na_2CO_3$, 35 mM/L $NaHCO_3$, pH 9.6). 100 μL of the antigen solution was added to each well of a 96-well plate and coated overnight at 4°C. On the following day, the CBS was removed. 150 μL of PBST (137 mM/L NaCl, 2.7 mM/L KCl, 10 mM/L $Na_2HPO_4$, 2 mM/L $KH_2PO_4$, 0.05% Tween 20, pH 7.4) was added to each well and washed on a shaker for 5 min, repeating this process 5 times. Then, 150 μL of 1% BSA was added to each well and blocked at room temperature for 2 h. The blocking buffer (1% BSA) was removed. Each well was then filled with 200 μL of PBST and washed on a shaker for 5 min, repeated for a total of 5 times. Subsequently, 100 μL of rabbit serum, diluted with blocking buffer (starting at a 1:500 dilution and performing serial dilutions), was added to each well and incubated at 25 °C for 1.5 h. Rabbit negative serum was used as a control. The primary antibody was removed. Each well was then filled with 200 μL of PBST and washed on a shaker for 5 min, repeating this process five times. Subsequently, 100 μL of HRP-conjugated goat anti-rabbit IgG (diluted 1:2000 with blocking buffer) was added to each well and incubated at 37 °C for 45 min. The secondary antibody was removed. Each well was then filled with 150 μL of PBST and washed on a shaker for 5 min, repeating this process five times. Subsequently, 100 μL of TMB substrate was added to each well and incubated at 37 °C for 15 min. 50 μL of 2 mol/L $H_2SO_4$ was quickly added to each well to terminate the color development. The absorbance at 450 nm was measured using a microplate reader. The antibody titer results were analyzed, with a positive result defined as a P/N ratio greater than 2.1.

## 2.8. Immunohistochemistry

Operation of IHC according to the method previously described [30]. The experimental procedure can be summarized as follows: First, the sections were dewaxed and rehydrated using xylene and ethanol, followed by washing with PBS. An endogenous peroxidase blocker was then applied. After PBS washing, antigen retrieval was performed using EDTA. Then, the procedure was carried out according to the instructions of the UltraSensitive SP IHC Kit (MaxVision, China). Polyclonal antibody of anti-Annexin B1/Annexin B2 (1:2000) was incubated at 4 °C for about 14 h. The sections were then incubated with biotin-labeled goat anti-rabbit IgG secondary antibody at room temperature for 1 h. DAB chromogenic solution was added for staining, followed by counterstaining with hematoxylin and differentiation with hydrochloric acid alcohol.

## 2.9. Prothrombin time (PT) and activated partial thromboplastin time (APTT) of Annexin B1 and B2

The venous blood of 10 healthy human volunteers was collected using sodium citrate-coated blood collection tubes, and plasma was subsequently isolated by centrifugation. Each sample was tested individually, rather than pooling 10 samples for detection. Recombinant Annexin B1 and

B2 were added to plasma to achieve final protein concentrations of 5 μg/mL, 10 μg/mL, 20 μg/mL, 40 μg/mL, 50 μg/mL, 100 μg/mL, 125μg/mL, 150μg/mL, 175μg/mL, and 200 μg/mL, respectively. The original plasma, BSA (200 μg/mL), and the plasma supplemented with normal saline (NS) were used as controls (The volume of normal saline added should be based on the volume of protein solution used to prepare the 200 μg/mL annexin plasma). The APTT and PT measurements were performed according to the instructions provided in the APTT Assay Kit (Zhongtai Biotech, China) and the PT Assay Kit (Zhongtai Biotech, China), respectively. The instrument used for the measurement was an XN06-II semi-automatic coagulometer (WuHan King Diagnostic Technology CO., LTD.). The reference normal range for APTT is 24-36s. The reference normal range for PT is 11-15s. Each sample assay was technically replicated three times. Statistical analysis of the results was performed using one-way analysis of variance (ANOVA).

## 2.10. Liposome binding assay

We adopted procedure A from the published literature [31]. Each spot contained 5 μg of lipids, and the proportions were as follows: DPPS/DPPC - 5/0 μg, 4/1 μg, 3/2 μg, 2/3 μg, 1/4 μg, and 0/5 μg. In addition, we made the following changes: 1mM $CaCl_2$ was added to the blocking solution.

## 2.11. Culture and transfection of C2C12

C2C12 cells (mouse myoblasts) were cultured in basal medium (DMEM includes 1% Penicillin-Streptomycin Solution and 10% FBS) in a cell culture incubator containing 5% $CO_2$ at 37°C. C2C12 were seeded into a 6-well plate at $0.5-1\times10^6$ cells/well, and cells with a growth density of 70–90% can be used for transfection. We used Lipofectamine 2000 to transfect the plasmid (pCDNA3.1-His-B1 and pCDNA3.1-His-B2). Please refer to the provided instructions for detailed methods. The culture medium was replaced with differentiation medium (High Glucose DMEM includes 1% Penicillin-Streptomycin Solution and 2% Horse serum) after 8 hours of transfection. Myotubes were obtained by cultivating C2C12 for three days in a differentiation medium. Extract total cell protein and conduct Western blot to verify transfection efficiency.

## 2.12. Plasma membrane repair kinetics upon laser injury in live cells

For specific details of the test, refer to Protocol A in the literature [32]. Use a 405nm ultraviolet ablation laser to irradiate a 1–2 μm circular region with the following settings: 100% power, damage time 40s. Use a 60× objective to collect images. Cells were imaged three times before injury and once every 10 s within 6 minutes after injury (excitation wavelength 510 nm; emission wavelength 620 nm). FM1–43 dye concentration was 1.6 μM. Repeat steps for at least 15 cells per condition, obtained in three independent experiments. Representative images for each condition are selected at different time points starting before injury, to assemble a panel of images showing the influx of FM dye under the different conditions. The change of fluorescence intensity (F/F0) in the whole process was calculated by measuring the fluorescence intensity of FM dye in the whole cell (F: the fluorescence intensity at each time point; F0: the fluorescence intensity before injury). Ensure that the microscope (Olympus FV3000, Japan) settings are consistent for each run of the trauma detection experiment. For statistical analysis, area under the curve (AUC) analysis followed by unpaired Welch's t-test are calculated.

## 2.13. Data analysis

The results were expressed as means ± SD. SPSS 22 statistical software was used for statistical analysis.

## 3. Result

### 3.1. Expression, purification, and western blotting of Annexin B1 and B2

Fig 1A and 1B showed that pET-28a-Annexin B1 and pET-28a-Annexin B2 appeared target bands around 35 KDa after induction by IPTG culture at 20 °C and 37 °C, and both proteins were expressed in the supernatant and precipitation. The protein expression in the precipitation was significantly higher than that in the supernatant at both temperatures, and the protein expression at 20 °C was higher than that at 37 °C. We chose the protein expressed in the supernatant at 20 °C for follow-up purification due to the lack of biological activity of the recombinant protein in the inclusion body. The supernatant proteins were purified with different gradient concentrations of imidazole after a large number of proteins were expressed at 20 °C. Fig 1C and 1D showed Annexin B1 and Annexin B2 with high purity and fewer impurity bands through elution with 500 mM imidazole.

The molecular mass of Annexin B1 was about 40 kDa and B2 was about 43 kDa (Fig 1E). Western blot results showed that B1 and B2 could react with the positive serum of *Cysticercus cellulosae* and anti-His monoclonal antibody (Fig 1F and 1H). But B1 and B2 did not react with pig negative serum (Fig 1G). This proves that the proteins have good reactivity.

### 3.2. Production of the polyclonal antibodies

Fig 2A and 2B showed that the polyclonal antibody can react with both proteins at the target position. But the rabbit negative serum does not react with both proteins, indicating that the polyclonal antibody has good specificity (Fig 2C). Fig 2D ELISA results showed that the negative rabbit serum did not react to the two antigens. The antibody titer was 0. The titer of polyclonal antibodies corresponding to the two proteins reached the highest level of 1:1024000 at 35 days.

### 3.3. Immunolocalization of annexin in *Cysticercus cellulosae*

Fig 3 showed that the tissue localization of Annexin B1 in *Cysticercus cellulosae* observed under objective lenses of 4×, 10×, and 20× magnification. It was observed that specific brown color appeared on the body surface of *Cysticercus cellulosae*. No specific staining was observed in the negative control. This showed that B1 was mainly located on the surface of *Cysticercus cellulosae* and digestive glands. Annexin B2 was also located on the body surface of *Cysticercus cellulosae*, as shown in Fig 3.

### 3.4. Anticoagulant activity

As shown in Fig 4A, when the concentration of Annexin B1 in plasma was less than 100 μg/mL, there were no significant difference among the groups. The clotting time remained within the "dotted line" (normal range). Then, the coagulation time was gradually prolonged with the increase of B1 concentration. The difference among the groups was extremely significant. The average coagulation time reached 58.8s when B1 concentration reached 175 μg/mL. This was the first time that it has exceeded the normal range of APTT by more than 10s, which holds pathological significance. The coagulation time was also extended to the longest when the concentration reached 200 μg/mL. The average was 93.3s. However, there was no significant difference among the groups when the concentration of Annexin B2 in plasma was less than 50 μg/mL. The coagulation time remained within the "dotted line" (Fig 4B). Then, the coagulation time was gradually prolonged with the increase of B2 concentration. The difference among the groups was extremely significant. The average coagulation time reached 53.7s when B2 concentration reached 150 μg/mL. The coagulation time was also extended

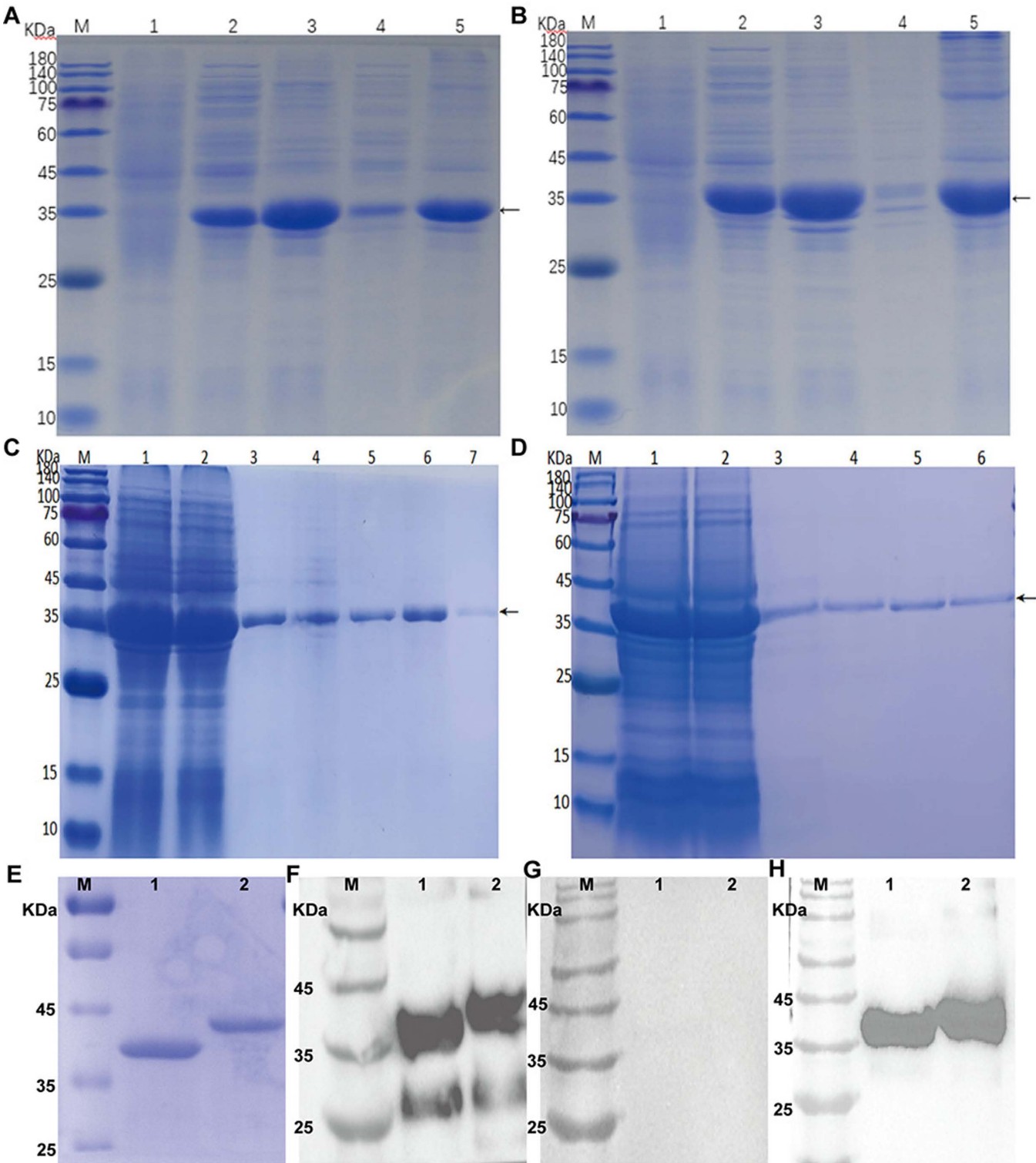

**Fig 1. Expression, purification, and western blotting of Annexin B1 and B2.** (A). SDS-PAGE analysis of recombinant B1 protein expression. (B). SDS-PAGE analysis of recombinant B2 protein expression. M: Protein Marker; 1: Total protein before induction; 2: 20°C supernatant; 3: 20°C precipitation; 4: 37°C supernatant; 5: 37°C precipitation. (C). Purification of Annexin B1 by nickel agarose affinity chromatography and SDS-PAGE analysis. M: Protein marker; 1: sample; 2: outflow; 3-4: 20 mM imidazole elution component; 5: 50 mM imidazole elution component; 6: 100 mM imidazole elution component; 7: 500 mM imidazole elution component. (D). Purification of Annexin B2 by nickel agarose affinity chromatography and SDS-PAGE analysis. M: Protein

marker; 1: sample; 2: outflow; 3: 20 mM imidazole elution component; 4: 50 mM imidazole elution component; 5-6: 500 mM imidazole elution component. (E). SDS-PAGE analysis of recombinant protein. M: Protein marker; 1: Annexin B1; 2: Annexin B2. (F). Recombinant protein with pig positive serum of *Cysticercus cellulosae*. M: Protein marker; 1: Annexin B1; 2: Annexin B2. (G). Recombinant protein with pig negative serum. (H). Recombinant protein with anti-His monoclonal antibody.

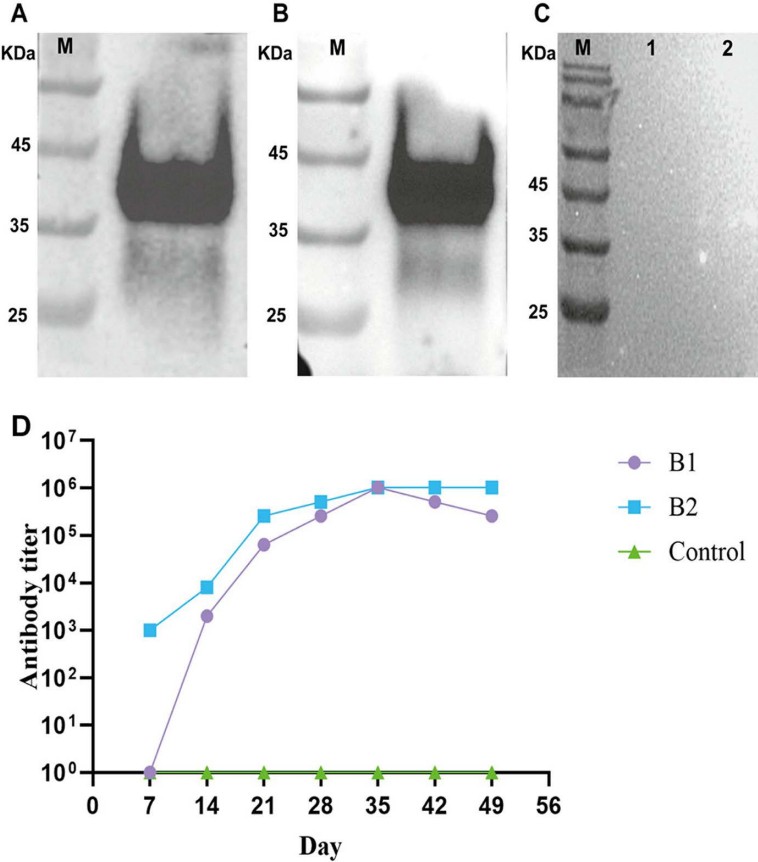

**Fig 2. Western Blotting analysis and titer determination of polyclonal antibodies.** (A). Annexin B1 with B1 polyclonal antibodies. (B). Annexin B2 with polyclonal antibodies. (C) Proteins with rabbit negative serum. M: Protein marke 1: Annexin B1; 2: Annexin B2. (D). Determination of rabbit polyclonal antibody titer.

to the longest when the concentration reached 200 µg/mL. The average was 116.4s. Interestingly, individual experimental plasma reach "non-coagulation" state when 200 µg/mL of B2 was added (XN06 semi-automatic hemagglutination instrument no longer measures the plasma that has not coagulated within 120s, and the judgment result indicates that it is not coagulated). Annexin B2 has a stronger effect on the coagulation activity of the endogenous coagulation system than Annexin B1.

There was no significant difference among the groups when the concentration of Annexin B1 in plasma was less than 125 µg/mL (Fig 4C). The clotting time remained within the "dotted line". Then, the coagulation time was gradually prolonged with the increase of B1 concentration. The difference among the groups was extremely significant. The average coagulation time reached 25.2s when B1 concentration reached 175µg/mL. This was the first time that it has exceeded the normal range of PT by more than 3s, which holds pathological significance.

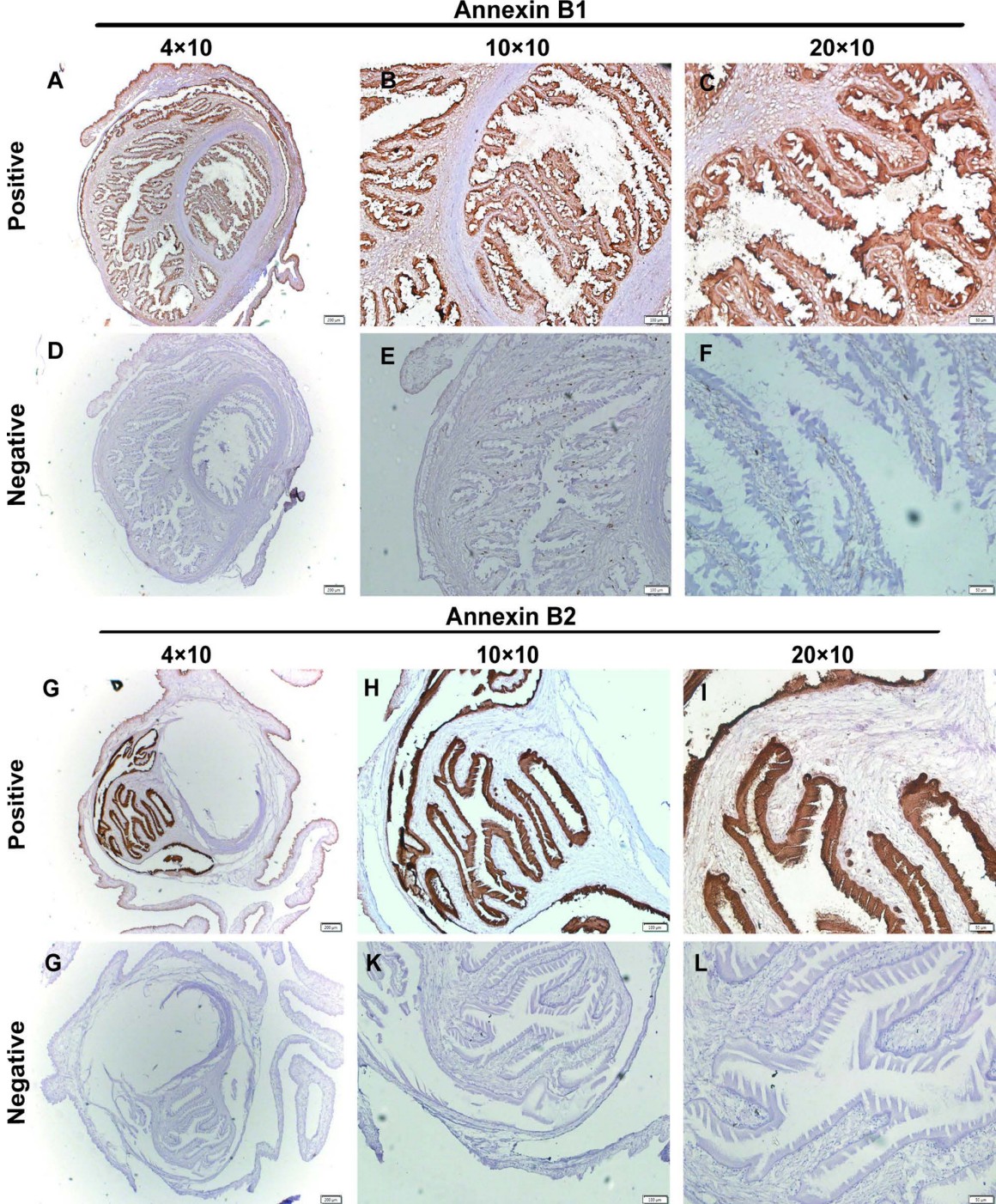

**Fig 3. Annexin B1 and Annexin B2 localization in *Cysticercus cellulosae*.**

The coagulation time was also extended to the longest when the concentration reached 200 μg/mL. The average was 35.6s. However, there was no significant difference among the groups when the concentration of Annexin B2 in plasma was less than 50 μg/mL. The coagulation time remained within the "dotted line", see Fig 4D. Then, the coagulation time was gradually

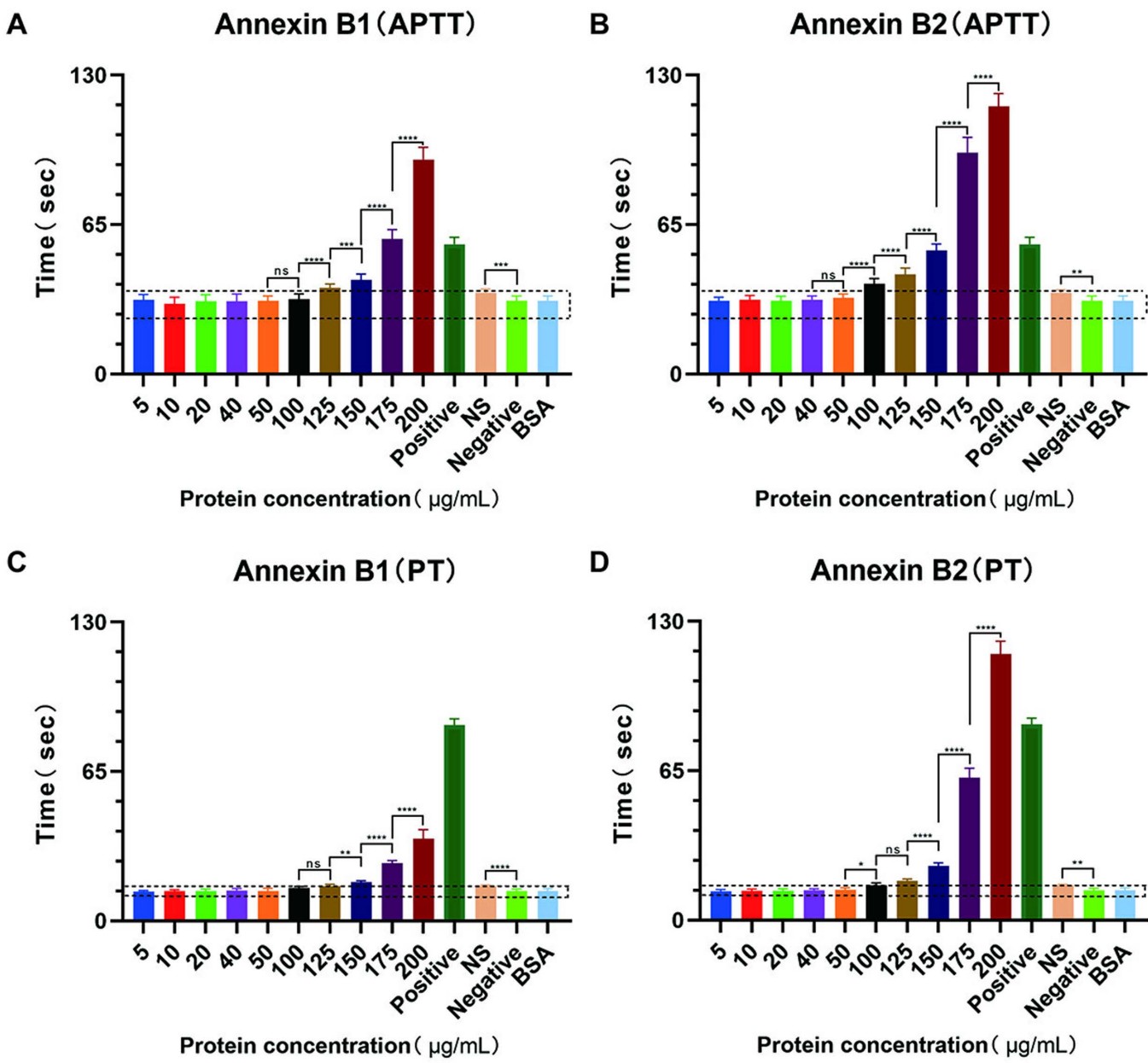

**Fig 4. Annexin B1 and B2 affect the determination of APTT and PT.** The dotted area is the normal time range. Plasma added with aprotinin as positive control of APTT. Plasma lacking coagulation factor FVII as positive control of PT. The original plasma samples were used as negative controls. NS is normol saline. $P<0.05$ (*), $P<0.01$ (**), $P<0.001$ (***), $P<0.0001$ (****), ns is non-significant (n=30).

prolonged with the increase of B2 concentration. The difference among the groups was extremely significant. The average coagulation time reached 23.6s when B2 concentration reached 150μg/mL.The coagulation time was also extended to the longest when the concentration reached 200 μg/mL. The average was 115.8s. Similar, individual experimental plasma reach "non-coagulation" state when 200 μg/mL of B2 was added. Annexin B2 has a stronger effect on the coagulation activity of the exogenous coagulation system than Annexin B1.

It was found that the coagulation time of the normal saline (NS) group was significantly higher than that of the normal plasma group by analyzing the data from the normal saline

group and the normal plasma group. This indicates that plasma diluted to a certain extent does indeed prolong the coagulation time. However, the prolongation effect was far lower than that of the groups with the same volume of B1 and B2, which excludes the reason that the two coagulation pathways are seriously interfered by the dilution of plasma. Additionally, in the four experimental groups, the addition of 200 μg/mL BSA to the plasma did not prolong the clotting time. This indicates that the protein concentration did not interfere with the coagulation process.

### 3.5. Liposome binding assays

The results of Fig 5A showed that Annexin B1 almost completely binds to No.1 spot, appearing completely black. Then, the proportion of spotted black became lower and lower when the proportion of DPPC increased gradually. There was a vague black imprint in No. 5 spot, but the black almost disappeared in No. 6 spot. The result of Fig 5B was similar to that of Fig 5A. But the difference was that at spot 4. The black imprint was not obvious. While at spots 5 and 6, black almost disappears.

This showed that B1 and B2 prefer to bind DPPS rather than DPPC in the presence of 1mM $CaCl_2$, and the binding ability of B2 may be weaker than that of B1.

### 3.6. Detection of transfection efficiency by western blot

The results of Fig 6E showed that the target bands appeared successfully and were thicker on No. 1 and No. 2. No specific bands were observed in lanes No. 3 and No. 4, indicating that the pCDNA3.1-His-B1 and pCDNA3.1-His-B2 were successfully transfected into C2C12 with high transfection efficiency.

### 3.7. Plasma membrane wounding and repair assays

C2C12 were exposed to injury by ablation laser, and membrane integrity was measured with FM1–43 dye in order to determine the kinetics of membrane wound healing. Weak fluorescence was observed on the plasma membrane before injury (-10s) in the negative control group ($+Ca^{2+}$). Then laser damage (0s) was performed on the part pointed by the arrow. The obvious membrane defect could be seen under intense laser stimulation, and the weak fluorescence (0s) absorbed on the plasma membrane was quenched. The fluorescence that flows into

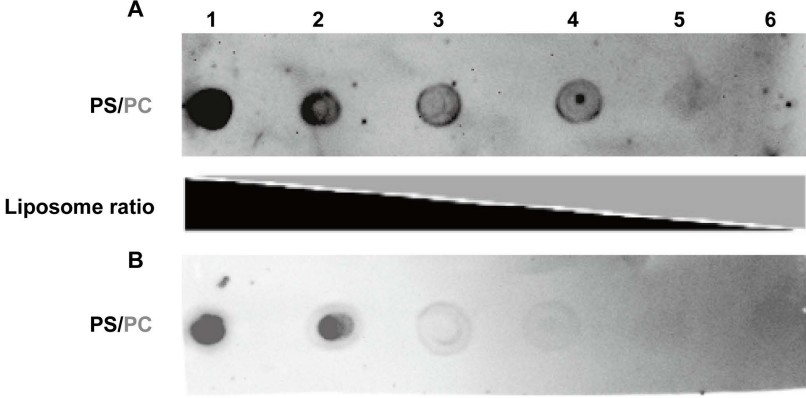

**Fig 5. Western blotting of liposome binding assay and total cellular protein.** Binding test of liposomes with different PS/PC combinations of recombinant Annexin B1 and B2. A: Annexin B1; B: Annexin B2. Numbers 1-6 represent different PS/PC binding ratio spots, which are 5/0, 4/1, 3/2, 2/3, 1/4 and 0/5 respectively.

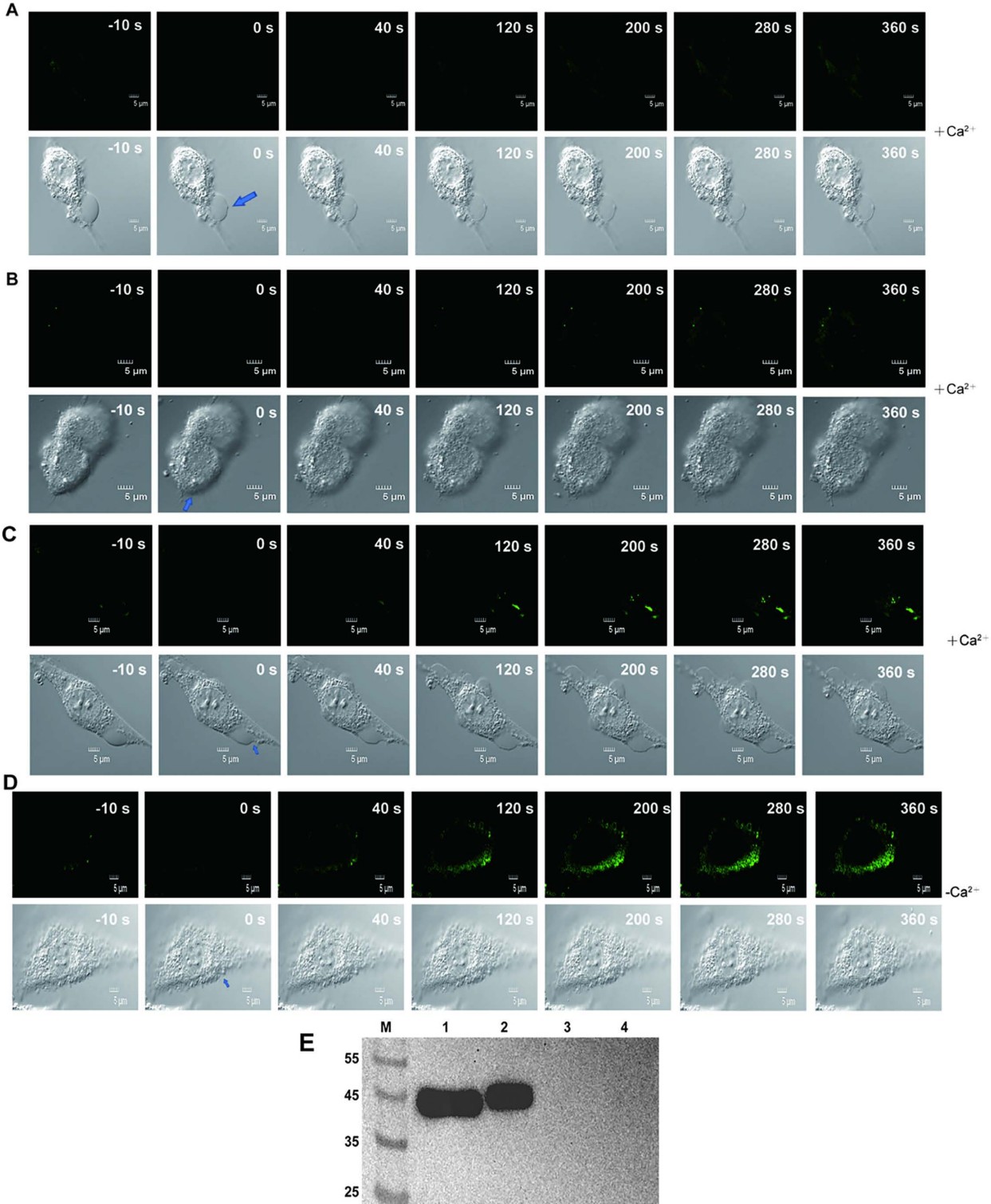

**Fig 6. Uptake of FM1-43 dye after laser injury in media in a C2C12 cell.** (A). +Ca²⁺ repair group of Annexin B1. (B). +Ca²⁺ repair group of Annexin B2. (C). +Ca²⁺ repair group of ordinary cells. (D). -Ca²⁺ no repair group of ordinary cells. The point of the arrow is the site of the injury. (E). Results of Western blot analysis of total cell protein. M: marker; 1: Cells transfected with Annexin B1 gene; 2: Cells transfected with Annexin B2 gene; 3: Cells transfected with pCDNA3.1-His; 4: Original C2C12.

the cells and adsorbs on the plasma membrane becomes brighter over time, see Fig 6C. The Fig 7C showed that the membrane wound seems to be roughly repaired around 300s, because the rate of fluorescence increase slows down. Fig 6D showed that the dye continued to flow into the cell after laser damage caused an obivious defect on the membrane in the negative control group ($-Ca^{2+}$). The results in Fig 7C confirm that the dye continues to flow into the cell with no sign of slowing down, indicating no plasma membrane repair. An additional movie file shows this in more detail [see S1 Video–S4 Video].

Fig 6A and 6B showed the plasma membrane repair of cells transfected with Annexin B1 and B2 genes after laser injury. Compared with the image at the same time point in Fig 6A and 6B, the fluorescence intensity of the image after 0s was significantly darker than that of the untransfected cells. The time period of membrane repair was significantly earlier than that of ordinary cells, as evidenced by the comparison of Fig 7A, 7B, and 7C. It is speculated that B1 and B2 play a key role in the early repair of the membrane. The comparative results of Fig 7C and 7D showed that the dye influx of the two groups in the pCDNA3.1-His group was consistent with that of the control group, which ruled out the effect of transfection on plasma membrane repair. There were significant differences in AUC between the groups with and

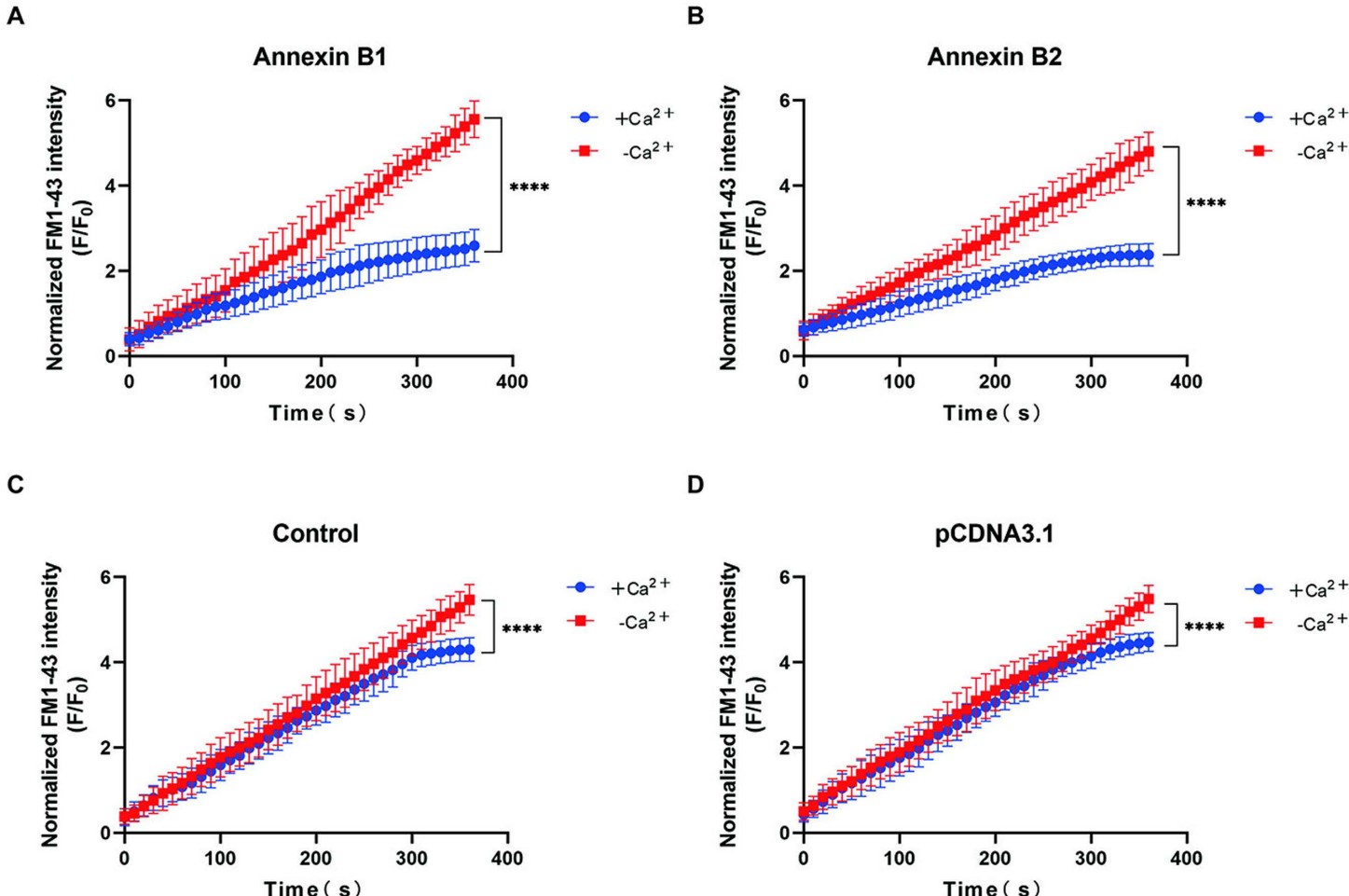

**Fig 7. Fluorescence intensity quantization of FM1-43 in cells.** (A). Group transfected with Annexin B1 gene. (B). Group transfected with Annexin B2 gene. (C). Group of original C2C12. (D). Group transfected with pCDNA3.1-His.

without Ca$^{2+}$ in each group, indicating that Ca$^{2+}$ indeed initiated the process of plasma membrane repair. An additional movie file shows this in more detail [see S5 Video–S8 Video].

## 4. Discussion

In this study, we described the key roles of Annexin B1 and B2 in the context of immune evasion mediated by *Cysticercus cellulosae* during invasion. This study conducted expression and purification, antigen tissue localization, coagulation function determination, phospholipid binding activity determination, and plasma membrane repair detection of the two annexins. Finally, the crucial role of two annexins in the invasion process has been discovered.

Most of the annexins were cytoplasmic or cytoskeletal proteins, but trace amounts of annexins (such as human Annexin A1, A2, and A5), which were also found in the extracellular region [33–36]. Although these annexins lacked the signal peptide sequence needed for exocrine secretion, this interesting phenomenon indicated that some annexins without signal peptides were also secreted. Anx B30 of *Clonorchis sinensis* was a secretory annexin that played a role in the interaction between the parasite and the host, influencing the host's autoimmune response [22]. Annexins were also found in the excretory-secretory products of *Echinococcus granulosus* and the cyst fluid. The sequence characteristics of B1 and B2 were analyzed using bioinformatics tools [37]. The results showed that the proteins were stable, lack transmembrane domains, and contain four classical Annexin domains. Interestingly, the probability that they contain signal peptides was very low, but we screened these two proteins from the proteomics of *Cysticercus cellulosae* in previous work, which illustrated that they were probably secreted proteins without signal peptides.

At present, the Annexin of many parasites have been histologically localized, and *Taenia multiceps* annexin B2 and B3 were expressed on the cyst wall [38]. The annexins B22, B30, B5a, B7a, and B5b of *Schistosoma mansoni* were all located on its tegument and can be recognized by positive mouse serum [39]. These annexins located in the tegument of the worm were presumed to play an important role in maintaining their structural integrity. Technically, the *Schistosoma* tegument was the main barrier to protect it. If tegument could be destroyed by blocking the function of the molecule (for example annexin), then a new control strategy might be discovered. The detection of the molecular distribution of Annexin in the muscle region of *Schistosoma mansoni* showed that they may be involved in the movement of parasites, which strongly supported that Annexin may played a role in the dynamic maintenance of related tegument through calcium regulation [40]. In addition, annexins B5a, B7a, and B5b were also detected in the intestinal wall of *Schistosoma japonicum* [39]. The intestinal wall was a stratified epithelium that participates in the digestion and absorption of undigested nutrients. These data suggested that Annexin B1 and B2 of *Cysticercus cellulosae* might play a potential role in maintaining the integrity of the worm, assisting in movement, and absorbing and digesting nutrients.

Previous studies have shown that in the APTT test, when the concentration of Annexin B1 reached 125 µg/mL, the blood reached a state of "non-coagulation"[24]. Another study showed that Annexin B1 did not affect PT, and Annexin B2 did not affect APTT [25]. However, our experimental results showed that when the concentration of Annexin B1 reached 200 µg/mL, the clotting time of APTT was obviously prolonged, but the clotting did not reach the state of "non-coagulation". Moreover, the two kinds of Annexin have obvious effects on APTT and PT when they reach a certain concentration, and the effect of Annexin B2 was stronger. The annexin of *Schistosoma bovis* also has anticoagulant properties [41]. It has been observed that adult *Schistosoma bovis* activates the fibrinolytic system to prevent peripheral thrombosis, specifically through the protein receptor of plasminogen expressed on the surface

of its membrane, one of which has been identified as enolase [41]. Human Annexin A2 and A5 also have anticoagulant activity A2 interacted with tissue plasminogen activator through its N-terminal LCKLSL motif [42]. Extracellular Annexin A5 showed anticoagulant activity dependent on its calcium regulatory binding to anionic phospholipids (exposed to activated platelets or endothelial cells) [33]. The platelet biofilm was a typical phospholipid bilayer structure. The neutral phospholipids in the outer layer were dominant, such as phosphatidylcholine and sphingomyelin, while the anionic phospholipids in the inner layer are dominant, such as phosphatidylserine, phosphatidylinositol, and phosphatidylethanolamine [43]. Under normal physiological conditions, blood remains in a liquid state. When platelets were stimulated by collagen or thrombin, the sphingomyelin and phosphatidylcholine on the outside of the platelet membrane were reversed with the medial phosphatidylethanolamine and phosphatidylserine, thus increasing the amount of phosphatidylethanolamine and phosphatidylserine on the surface of the membrane [44]. They provide the surface for the coagulation complex enzymes during the coagulation process and protect thrombin from being inactivated by plasma and cellular serine protease inhibitors to prevent disseminated intravascular coagulation [44]. Due to the phospholipid binding activity of B1 and B2, we speculated that they competitively bind to phospholipids, such as phosphatidylserine, in the presence of calcium ions. This interference disrupts the formation of some activated coagulation complexes, thereby inhibiting both endogenous and exogenous coagulation pathways. Therefore, it was speculated that Annexin B1 and B2 played an anticoagulant role in the process of *Cysticercus cellulosae,* penetrating through blood vessels and migrating to subcutaneous tissue, myocardium, and skeletal muscle.

The plasma membrane of eukaryotic cells was composed of phospholipid bilayers with integrated transmembrane proteins, which basically formed a physical barrier to separate the cell from the extracellular environment and maintain a basic osmotic gradient outward [45]. During the lifespan of most cells, the integrity of the plasma membrane was often damaged in different ways. The plasma membrane of cells in mechanically active tissue environments (such as muscle and lung cells) was often damaged, posing a direct threat to cell survival if not repaired. Therefore, cells have developed an effective plasma membrane repair mechanism to deal with membrane damage and ensure the stability of the intracellular environment. The repair mechanism was strictly dependent on the entry of extracellular calcium ($Ca^{2+}$) into the cell through the wound and involves several cellular processes, including cytoskeleton reorganization, exocytosis, endocytosis, and membrane exfoliation [46–51]. There have been many studies on the role of annexin in plasma membrane repair in vertebrates. The selection of damage type was very important, and many methods of damage creation were difficult to repeat accurately in the study of repair. This study selected a method to monitor the dynamics of membrane repair in living cells after UV ablation laser-induced plasma membrane damage. It can produce local, repeatable, non-fatal damage, and combined with live-cell imaging to track fluorescently labeled proteins during repair. This approach was widely used in various studies of mammals and invertebrates [52–54]. During the experiment, some cells did not immediately exhibit membrane damage after laser damage, but produced vesicular protrusions, which usually appeared in other parts of the cell (other than the site of direct laser stimulation) in a repetitive and asynchronous manner. Bubbles were a common phenomenon in animal cells and observed during cell division, apoptosis, and some types of cell migration. Bubbles were primarily a physical process rather than a chemical one, occurring on a scale of time and length. During this process, a small portion of the bilayer membrane separated from the cortex. It grown into a bubble if it reached a critical size. This detachment was caused by local contraction of the actin cortex driven by myosin II, and the bubble size depends on membrane tension, cytosol pressure, and adhesion between the membrane and the actin

cortex [55]. Bubbles on cells have been proposed as a possible mechanism to resist plasma membrane damage by trapping damaged membrane segments and adjacent cytoplasm. These damaged components are then isolated in vitro by ANX A1 and membrane-binding proteins, limiting the loss of intracellular components to the area of the bubble [56]. The studies have shown that damaged cells without blistering are more likely to die, while blistered damaged cells are more likely to recover [56]. Whether B1 and B2 could increase the frequency of bubbles in damaged cells or participate in a certain process of bubble formation remains to be studied.

The example of real-time imaging of cell membrane repair in response to laser injury, as presented in Sønder SL's article, shows a two-fold difference in F/F0 between control cells with and without $Ca^{2+}$ [32]. However, the results observed in Fig 7C are different from those described above. Additionally, in Fig 7A and 7B, the fluorescence intensity in the $+Ca^{2+}$ group shows a linear accumulation rather than a saturation curve. In Sønder SL's protocol, the microscope settings used were '2.6% power, 200 Hz repetition rate, pulse energy >60 μJ, and pulse length <4 ns', with a damage duration of 4 ns [32]. However, during our experiments, we found that these parameters were not applicable to our setup (our microscope could only control power). We ultimately used a power setting of 100% with a damage duration of 40 s. We believe that the differences in microscope parameters and damage duration between our study and the reference led to inconsistencies in the degree of damage inflicted. This, in turn, increased the difficulty of cell repair, resulting in delayed repair time and extent. Consequently, these factors contributed to the issues we previously described.

In this study, during the process of membrane resealing, electron microscope images revealed that the plasma membrane appeared to extend "filaments" to repair the membrane defect (Fig 6A and 6B). This was not the first time that this situation has been found in the study. In the human skeletal muscle cell membrane repair model, it was proposed that $Ca^{2+}$ influx induces annexins (especially ANX A5 and ANX A6) to recruit from the plasma membrane, followed by actin depolymerization and lysosome exocytosis to reduce membrane tension [57]. The vesicles in the cell were recruited to the damaged site and gather to form a "patch" that blocks the rupture. ANX A6 and ANX A5 induced sarcolemmal elongation and folding to form a tight structure. At the same time, the elongation of the sarcolemma and the recruitment of intracellular vesicles might lead to the accumulation of ANX, which leads to the folding and bending of excess sarcolemma and the formation of the cap subdomain [58]. It was speculated that the "filament" observed was the elongated and folded sarcolemma. In this study, it was preliminarily determined that *Cysticercus cellulosae* Annexin B1 and B2 could help to repair the plasma membrane. It was speculated that when the host immune system attacked the worm, these two proteins were involved in repairing the worm's body surface to resist the immune system's attack. However, due to the complexity of the plasma membrane repair mechanism, the cellular process through which they repair the plasma membrane and how they interact with other proteins or phospholipids still need to be studied.

In this study, we conducted preliminary tissue localization of Annexin B1 and B2, confirming their anticoagulant and plasma membrane repair functions and furthermore revealed that these two proteins played a crucial role in the migration and invasion of the host. It was preliminarily judged that it might help maintain the structural integrity of the worm through plasma membrane repair. This finding offered a new perspective for developing drugs against *Cysticercus cellulosae* and for early prevention and treatment. However, it was unknown whether the two proteins can participate in immune escape by regulating the chemotaxis and proliferation of immune cells.

## 5. Conclusions

In this study, Annexin B1 and B2 of *Cysticercus cellulosae* were successfully expressed and purified. It was found that they were located on the surface of the body and the surface of the internal digestive gland. Both proteins would prolong the time of the endogenous and exogenous coagulation pathways upon reaching a certain concentration, and both have a preference for binding to PS rather than PC. C2C12 cells transfected with the B1 and B2 genes took less time to repair the plasma membrane after laser-induced damage. This discovery provided a new perspective and valuable clues for a deeper understanding of the invasion mechanism of *Cysticercus cellulosae*.

## Supporting information

**S1 Video. +Ca$^{2+}$ repair group of ordinary cells (Fluorescence view).**
(MP4)

**S2 Video. +Ca$^{2+}$ repair group of ordinary cells (Bright - field view).**
(MP4)

**S3 Video. -Ca$^{2+}$ no repair group of ordinary cells (Fluorescence view).**
(MP4)

**S4 Video. -Ca$^{2+}$ no repair group of ordinary cells (Bright - field view).**
(MP4)

**S5 Video. +Ca$^{2+}$ repair group of Annexin B1 (Fluorescence view).**
(MP4)

**S6 Video. +Ca$^{2+}$ repair group of Annexin B1 (Bright - field view).**
(MP4)

**S7 Video. +Ca$^{2+}$ repair group of Annexin B2 (Fluorescence view).**
(MP4)

**S8 Video. +Ca$^{2+}$ repair group of Annexin B2 (Bright - field view).**
(MP4)

**S1 Data. The values used to build figures.**
(XLSX)

## Author contributions

**Conceptualization:** Xiaolei Liu, Shumin Sun.

**Data curation:** Mengqi Wang, Rui Duan.

**Formal analysis:** Mengqi Wang, Rui Duan.

**Funding acquisition:** Xiaolei Liu, Shumin Sun.

**Investigation:** Peixia He, Dejia Zhang.

**Methodology:** Xiaolei Liu, Shumin Sun.

**Resources:** Xiaolei Liu, Shumin Sun.

**Supervision:** Xing Yang.

**Validation:** Mengqi Wang, Rui Duan, Yuyuan Zhao, Sirui Wang.

**Visualization:** Mengqi Wang, Rui Duan.

**Writing – original draft:** Peixia He, Dejia Zhang.

**Writing – review & editing:** Xiaolei Liu, Shumin Sun.

## Acknowledgments

We thank the State Key Laboratory for Diagnosis and Treatment of Severe Zoonotic Infectious Diseases, Jilin University, for providing experimental platforms. We would also like to thank all those who contributed to this work.

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
