## [Decision Letter · Decision Letter 0]

16 Feb 2025

PNTD-D-24-01289

Functional Identification of Annexin B1 and Annexin B2 from Cysticercus cellulosae and Their Mechanism in Plasma Membrane Repair

Dear Dr. Sun,

Thank you for submitting your manuscript to PLOS Neglected Tropical Diseases. After careful consideration, we feel that it has merit but does not fully meet PLOS Neglected Tropical Diseases's publication criteria as it currently stands. Therefore, we invite you to submit a revised version of the manuscript that addresses the points raised during the review process.

Please submit your revised manuscript within 60 days Apr 17 2025 11:59PM. If you will need more time than this to complete your revisions, please reply to this message or contact the journal office at plosntds@plos.org. Please include the following items when submitting your revised manuscript:

We look forward to receiving your revised manuscript.

Kind regards,

Richard A. Bowen, DVM PhD

Academic Editor

Jong-Yil Chai

Section Editor

Shaden Kamhawi

co-Editor-in-Chief

Paul Brindley

co-Editor-in-Chief

**Additional Editor Comments :**

Your manuscript has finally been evaluated by three - apologies for how low it took to get the reviews back. All three reviewers had some major criticisms. Pleease review their comments and prepare a revised version of the manuscript as detailed in this letter and also prepare a responses to reviewer document that outlines you change and reasoning behind the changes or defend not changing certain aspects of the paper.

**Journal Requirements:**

At this stage, the following Authors/Authors require contributions: Peixia He, Dejia Zhang, Mengqi Wang, Rui Duan, Yuyuan Zhao, Sirui Wang, Xing Yang, Xiaolei Liu, and Shumin Sun. Please ensure that the full contributions of each author are acknowledged in the "Add/Edit/Remove Authors" section of our submission form.

4) We note that your Data Availability Statement is currently as follows: "All data generated or analyzed during this study are included in this published article." Please confirm at this time whether or not your submission contains all raw data required to replicate the results of your study. Authors must share the “minimal data set” for their submission. PLOS defines the minimal data set to consist of the data required to replicate all study findings reported in the article, as well as related metadata and methods (https://journals.plos.org/plosone/s/data-availability#loc-minimal-data-set-definition).

5) Please ensure that the funders and grant numbers match between the Financial Disclosure field and the Funding Information tab in your submission form. Note that the funders must be provided in the same order in both places as well. Currently, the order of the grants is different in both places.

**Comments to the Authors: **

**Please note that one of the reviews is uploaded as an attachment.**

**Reviewers' Comments:**

Reviewer's Responses to Questions

**Key Review Criteria Required for Acceptance?**

**Methods**

-Are the objectives of the study clearly articulated with a clear testable hypothesis stated?

-Is the study design appropriate to address the stated objectives?

-Is the population clearly described and appropriate for the hypothesis being tested?

-Is the sample size sufficient to ensure adequate power to address the hypothesis being tested?

-Were correct statistical analysis used to support conclusions?

-Are there concerns about ethical or regulatory requirements being met?

Reviewer #1: Yes, the objectives of the study clearly articulated with a clear testable hypothesis stated.

Yes, the study design appropriate to address the stated objectives.

Yes, the sample size is sufficient.

Yes, correct statistical analysis is used to support conclusions.

Reviewer #2: Line 158 Expression of Annexin B1 and Annexin B2

Plese provide genes' IDs or sequence used

Reviewer #3: (No Response)

**Results**

-Does the analysis presented match the analysis plan?

-Are the results clearly and completely presented?

-Are the figures (Tables, Images) of sufficient quality for clarity?

Reviewer #1: Yes, the analysis presented match the analysis plan.

Yes, the results clearly and completely presented.

Yes, the figures are of sufficent quality for clarity.

Reviewer #2: Figure 4, no scale bar.

Figure 5, provide raw data and details on statistical analysis. There are only two levels of significance either ns or P＜0.0001 (****) that seems possible although improbable.

Figure 8. Repair of plasma membrane after transfection. Should be Repair after the injury.

Figure 9. Fluorescence intensity quantization of FM1-43 in cells.

Example of real time imaging of cell membrane repair in response to laser injury from “[29] Sønder SL, Ebstrup ML, Dias C, Heitmann ASB, Nylandsted J. Plasma Membrane 658 Wounding and Repair Assays for Eukaryotic Cells. Bio Protoc. 2022;12(11):e4437. 659 doi:10.21769/BioProtoc.4437” has two-fold difference in F/F0 in control cells with and without Ca2+ because of the divalent-membrane interaction itself. Authors may want to discuss why there is no Ca2+ effect in control cells in their control experiment.

FM1-43 kinetics supposed to show saturation upon membrane repair. Authors can discuss why they see near linear dye accumulation.

Methods section was going to use (AUC) analysis followed by unpaired t-test with Welch’s that appears missing from Figure 9.

Reviewer #3: (No Response)

**Conclusions**

-Are the conclusions supported by the data presented?

-Are the limitations of analysis clearly described?

-Do the authors discuss how these data can be helpful to advance our understanding of the topic under study?

-Is public health relevance addressed?

Reviewer #1: Yes, the conclusions are supported by the data presented.

Yes, the limitations of analysis clearly described.

Yes, the authors discuss how these data can be helpful to advance our understanding of the topic under study.

Yes, the public health relevance is addressed.

Reviewer #2: Conclusions are supported by the data presented

Reviewer #3: (No Response)

**Editorial and Data Presentation Modifications?**

Reviewer #1: English writing is not fluent, requring major revision.

Reviewer #2: Authors may want to extend and clarify figures captions.

Reviewer #3: (No Response)

**Summary and General Comments**

Reviewer #1: This manuscript described the investigation on the functions of Annexin B1 and B2 from Cysticercus cellulosae, the larval stage of parasitic tape worm Taenia solium. This parasite can cause severe human disease by invading human brain, retina and heart. However, the invading mechanism is still unclear. Annexins are evolutionarily highly conserved, calcium-dependent phospholipid-binding proteins. In spite that previous studies have reported that annexins have various functions, including anti-inflammatory properties, maintaining fibrinolytic balance, anticoagulant, regulating vesicle transport, and controlling the formation of ion channels, little is known about the annexins from C. cellulosae. The present study investigated the localization of these two proteins in the worm, assessed their anticoagulant function, phospholipid binding activity and membrane repair function. The results revealed that the two proteins were located on the surface of the body and the surface of the internal digestive gland. Both proteins would prolong the time of the endogenous and exogenous coagulation pathways and both have a preference for binding to PS rather than PC. C2C12 cells transfected with the B1 and B2 genes took less time to repair the plasma membrane after laser-induced damage. The findings from this study added our knowledge about the functions of these two protein, which helped us to further understand the invasion mechanism of C. cellulosae. The aim of this study is clearly formulated in the introduction, the methods selected are appropriate and the results are reasonably presented. However, the English writing is not fluent, requiring substantial revision.

Some minor errors are listed as example:

Line 26, “coli” should be italic.

Line 160: “Escherichia coli” should be read as italic.

Line 179: “B” should be lower case.

Figures 3 and 4 can be combined into one figure.

Line 428, “+Ca2+ no repair group of Annexin B2.”, is this sentence correct?

Reviewer #2: Ms addresses important parasitic disease caused by Cysticercus cellulosae. Authors expressed and purified Annexins B1 and B2 in prokaryotic vector and synthesized antibodies. The technical achievement allowed to them to localize Annexins at the surface of the parasite as expected. Anticoagulant activity of the Annexin was measured and found to be extremely significant (P<0.0001) for some protein concentrations. The statistical test performed for the anticoagulant activity’s measurement requires additional transparency and possible reevaluation. Another discrepancy is a lack of Ca2+ effect on the cell membrane repair in control cells and lack of saturation of the FM1-43 dye implies lack of membrane repair but rather continuous dye uptake by the damaged cells. The new experiment author may want to consider is to replicate the FM1-43 dye protocol in control conditions or discuss the observed discrepancies.

Authors may discuss more Annexins as a drug target. Possible drugs classes targeting them, interaction with host proteins etc.

Reviewer #3: (No Response)

PLOS authors have the option to publish the peer review history of their article (what does this mean? ). If published, this will include your full peer review and any attached files.

**Do you want your identity to be public for this peer review?** For information about this choice, including consent withdrawal, please see our Privacy Policy .

Reviewer #1: No

Reviewer #2: No

Reviewer #3: No

**Figure resubmission:**
---

## [Editor Report · Decision Letter 1]

27 Mar 2025

Dear Mr Sun,

We are pleased to inform you that your manuscript 'Functional Identification of Annexin B1 and Annexin B2 from Cysticercus cellulosae and Their Mechanism in Plasma Membrane Repair' has been provisionally accepted for publication in PLOS Neglected Tropical Diseases.

Best regards,

Richard A. Bowen, DVM PhD

Academic Editor

Jong-Yil Chai

Section Editor

Shaden Kamhawi

co-Editor-in-Chief

Paul Brindley

co-Editor-in-Chief

Thannk your for the comprehensive and thoughtful response to reviewer comments. This manuscript will be a valuable contribution to the field.

---

## [Editor Report · Acceptance letter]

Dear Mr Sun,

We are delighted to inform you that your manuscript, "Functional Identification of Annexin B1 and Annexin B2 from Cysticercus cellulosae and Their Mechanism in Plasma Membrane Repair," has been formally accepted for publication in PLOS Neglected Tropical Diseases.

Best regards,

Shaden Kamhawi

co-Editor-in-Chief

Paul Brindley

co-Editor-in-Chief
